# Mechanistic and Functional Shades of Mucins and Associated Glycans in Colon Cancer

**DOI:** 10.3390/cancers12030649

**Published:** 2020-03-11

**Authors:** Ramesh Pothuraju, Shiv Ram Krishn, Shailendra K. Gautam, Priya Pai, Koelina Ganguly, Sanjib Chaudhary, Satyanarayana Rachagani, Sukhwinder Kaur, Surinder K. Batra

**Affiliations:** 1Department of Biochemistry and Molecular Biology, University of Nebraska Medical Center, Omaha, NE 68198, USA; ramesh.pothuraju@unmc.edu (R.P.); shiv.ram.krishn@jefferson.edu (S.R.K.); shailendra.gautam@unmc.edu (S.K.G.); ppai1@jhmi.edu (P.P.); koelina.ganguly@unmc.edu (K.G.); sanjib.chaudhary@unmc.edu (S.C.); srachagani@unmc.edu (S.R.); skaur@unmc.edu (S.K.); 2Fred and Pamela Buffett Cancer Center, University of Nebraska Medical Center, Omaha, NE 68105, USA; 3Eppley Institute for Research in Cancer and Allied Diseases, University of Nebraska Medical Center, Omaha, NE 68198, USA

**Keywords:** Mucins, MUC2, MUC5AC, MUC4, glycans, inflammatory bowel disease, colon and colorectal cancer

## Abstract

Mucus serves as the chief protective barrier against pathogenic and mechanical insults in respiratory, gastrointestinal, and urogenital tracts. Altered mucin expression, the major component of mucus, in conjunction with differential glycosylation has been strongly associated with both benign and malignant pathologies of colon. Mucins and their associated glycans arbitrate their impact sterically as well as mechanically by altering molecular and microbial spectrum during pathogenesis. Mucin expression in normal and pathological conditions is regulated by nonspecific (dietary factors and gut microbiota) and specific (epigenetic and transcriptional) modulators. Further, recent studies highlight the impact of altering mucin glycome (cancer-associated carbohydrate antigens including Tn, Sialyl-Tn, Sialyl-Lew A, and Sialyl-Lewis X) on host immunomodulation, antitumor immunity, as well as gut microbiota. In light of emerging literature, the present review article digs into the impact of structural organization and of expressional and glycosylation alteration of mucin family members on benign and malignant pathologies of colorectal cancer.

## 1. Introduction

Mucus contains water, immunoglobulins, and mucins. Alteration in its composition is associated with several cancers including colorectal cancer (CRC), and higher mucus secretion is a hallmark of colorectal carcinoma [1]. CRC is the third leading cause of cancer-related deaths in the United States with 147,950 and 53,200 estimated new cases and deaths, respectively, in 2020 [2]. The 5-year survival rate for localized CRC is around 89.8%. Most CRCs develop from the normal colon epithelium through sequential events leading from polyp-adenoma to carcinoma [3]. Moreover, CRC progression is associated with a sedentary lifestyle, environmental factors, and smoking, which damage the intestinal mucus barrier and alter expression of tumor suppressor genes and oncogenes [4,5]. CRCs that follow the polyp-adenoma-carcinoma pathway (accounting for the majority (70–80%) of CRC cases) typically possess 60% mutations in the adenomatous polyposis coli (*APC*) gene along with *KRAS* (45%) and *TP53* (54%) mutations. On the other hand, *BRAF* mutations are the most frequent (70–100%) in hypermutated microsatellite instability (MSI)-high (MSI-H) CRC cases that follow the serrated pathway and account for 20–30% of CRC (Figure 1) [6,7,8]. However, sessile serrated adenomas/polyps (SSA/Ps) are histologically different from hyperplastic polyps (HPs) and have increased risk for the CRC development [9]. Further, inflammatory bowel diseases viz. ulcerative colitis (UC) and Crohn’s disease (CD) also contribute towards the development of CRC [10]. Molecular alterations like MSI (15–30%) and like *BRAF* (~10%) and *KRAS* (~10–25%) mutations are linked with UC-associated CRC and contribute to damage of mucosal barriers, which further leads to UC-associated mucosal neoplasia. MSI-H and CpG island methylator phenotype (CIMP) molecular subtypes are the precursors of the serrated pathway, whereas MSI-H and microsatellite stability (MSS) are predictive of the conventional pathway [11,12]. Recently, CRC was divided into four consensus molecular subtypes (CMSs) with CMS1: hypermutated, MSI, and strong immune activation; CMS2: epithelial, marked WNT, and MYC signaling activation; CMS3: epithelial and evident metabolic dysregulation; and CMS4: prominent transforming growth factor beta (TGF) beta activation, stromal invasion, and angiogenesis [13].

Mucins are heavily glycosylated proteins in the mucus and are synthesized by the goblet cells of epithelial tissues in various organs such as the lungs, stomach, and intestine [14,15] and form ductal surfaces of liver, breast, pancreas, and kidney [16,17,18]. It plays an important role in hydration, lubrication, and protection of epithelial cell lining of ducts and airways from chemical and mechanical aggressions [19,20,21]. Mucins are categorized into membrane bound (MUC1, MUC3A/B, MUC4, MUC11-13, MUC15-17, MUC20, and MUC21), secretory (MUC2, MUC5AC, MUC5B, MUC6, and MUC19), and non-gel-forming (MUC7) groups. It contains tandem repeat regions which are rich in proline, threonine, and serine residues that are heavily O-glycosylated and less *N*-linked glycosylated [22]. Altered expression or glycosylation of mucins has been implicated in the development of CRC along with other cancers including breast, ovarian, and pancreatic cancers [22,23]. Mucin expression is regulated by both nonspecific (dietary factors and gut microbiota) [24,25,26,27] and specific (epigenetic and transcriptional) modulators [28,29,30,31]. As shown in Figure 1, differential expression of mucins and associated glycans are observed in both conventional and serrated pathways. Understanding the pathogenesis of mucins and their signaling is critical to target benign ailments and malignancies of the colon. Therefore, this review focuses on the structural organization, regulation, signaling, and expression of different mucins in both benign and malignant development of colon.

## 2. Colon Mucus Organization and Its Composition

The colon mucus layer possesses a distinctive organization and function. As shown in Figure 2, in contrast to the small intestine, colon mucus has two well-demarcated zones comprising an outer loosely arranged and an inner compact stacked lamellar MUC2 sheet layer which is similar to the stomach that has two layers of secretary mucin MUC5AC [32]. The outer layer harbors a large number of commensal microflora, while the inner layer is impenetrable to these microorganisms due to its more compact and dense organization. In addition, the outer layer can be easily aspirated while the inner layer is relatively adherent and is secreted by the goblet cells at the colonic surface [33]. Owing to the high water content (98%) of mucin, visualization of the mucin layer was a challenge until recently. However, it was discovered that charcoal and the use of the Carnoy fixative renders the mucus layer easily visible [34].

## 3. Glycosylation of Colonic Mucins

Mucins are both *N*- and *O*-glycosylated during posttranslational modification [35]. Each form of glycosylation occurs in a distinct cellular compartment and significantly alters the structure and function of the mucin. The *O*- and *N*-glycans on mucins have shown to regulate the structural and functional properties of mucins [35]. *N*-glycosylation occurs by the addition of a sugar moiety to an asparagine residue in the Endoplasmic Reticulum (ER) [36], while *O*-glycosylation occurs in the Golgi and is initiated by the addition of a sugar moiety at serine/threonine residues in the tandem repeat regions of mucins. Mucin-type *O*-glycosylation is the predominant type of glycosylation and is a highly complex phenomenon. The covalent addition of glycans to the polypeptide backbone of mucins is regulated by a myriad of enzymes, leading to the formation of several core carbohydrate structures. Thus far, eight core carbohydrate structures on mucins have been recognized [37]. The sections below highlight the critical roles played by colonic mucin glycans under physiological conditions and pathological consequences of aberrant glycosylation in several colonic diseases.

### 3.1. Physiological Significance of Mucin Glycans

The *O*-glycans emanating from the core polypeptide of mucins enhance the structural complexity of the colonic mucus layer and play significant roles in intestinal homeostasis. Mucin *O*-glycans determine the composition of the gut microbiota via their interaction with bacterial cognate receptor adhesins [38,39]. For instance, a diverse array of oligosaccharides on the predominant colonic mucin MUC2 governs the region-specific distribution of colonic flora. These carbohydrate moieties also serve as a bacterial nutritional source [40]. Subtle changes in mucin oligosaccharides can affect resident microbial species, altering host susceptibility to intestinal diseases. In addition to the mucosal *O*-glycosylation, mucin *N*-glycan profile also plays a crucial role in maintaining the integrity of mucosal barrier and disease susceptibility. The transgenic β-1,4-galactosyltransferase I (βGalT1) enzyme plays an essential role in *N*-glycosylation, catalyzing the addition of galactose to N-acetylgalactosamine of type 2 *N*-glycans. In addition, mice overexpressing βGalT1 have a decreased predisposition to dextran sodium sulfate (DSS)-induced colitis, owing to an elevated *Firmicutes*-to-*Bacteroidetes* ratio [41]. 

### 3.2. Differential Glycosylation of Mucins in Colonic Diseases

Alterations of mucin *O-* and *N-*linked glycosylation is observed in several colonic diseases. Early evidence of altered mucin glycosylation came from studies focused on investigating the carbohydrate composition of mucins. Decreased carbohydrate content was observed in CRC mucins compared to their nonneoplastic counterparts [42]. Aberrant mucin glycans comprise the non-sialylated and sialylated forms of truncated carbohydrate antigens (Tn and T) and Lewis carbohydrate structures (Lewis a and Lewis x) [43]. The increased expression of sialyl-Tn (sTn) and sialyl-Lewis a and x has been reported in CRC [44,45]. Of note, increased expression of *O*-linked glycans such as sTn, sLe^a^, and sLe^x^ has been observed on MUC1 in CRC. The differential glycosylation of colonic mucins is attributed to defects in key components of glycosylation machinery, which includes glycosyltransferases and sugar-nucleotide transporters [46]. Core-1 and core-3 carbohydrate structures predominate the colonic mucus layer, and the loss of either of these has been linked to colitis and CRC [47]. In addition, several mouse models of colitis lack key glycosyltransferases that are involved in the biosynthesis of *O*-glycans [48].

Mucin-associated carbohydrate antigens are implicated in tumor growth and metastasis. The importance of mucin *O*-glycans in metastasis was demonstrated by using the *O*-glycosylation inhibitor benzyl-alpha-N-acetylgalactosamine. Treatment of high mucin-producing colon cancer cells such as HM 7, HM 3, and LS LiM 6 with this inhibitor has a significant effect on decreasing metastasis [49]. Similarly, another study on benzyl-alpha-N-acetylgalactosamine-treated HM 7 cells demonstrated a significant decrease in matrix metalloproteinase activity and reduced invasion. Further, inhibitor-treated colon cancer cells (HM7) showed decreased binding to Endothelial leukocyte adhesion molecule-1 coated plates [50]. This is possibly due to the decrease in peripheral carbohydrate structures such as SLe^x^ and SLe^a^ that have been shown to play an important role in intravasation/extravasation through their interaction with ELAM on endothelial cells. Interestingly, these peripheral carbohydrate structures also serve as ligands for β-galactoside-binding proteins such as galectins, which have been shown to augment metastatic potential of cancer cells by regulating homotypic and heterotypic cellular interactions [51]. Galectins belong to the lectin family of carbohydrate-binding proteins, which also include other lectins such as selectins and Siglecs [52,53,54]. The significance of selectins and Siglecs in mediating tumor growth and metastasis is well documented in several malignancies including CRC [55]. Mucins are the preferential substrates for these carbohydrate-binding receptors due to an abundance of *O*-glycans on their surface [56]. In conclusion, mucins are aberrantly glycosylated in benign and malignant pathologies of the colon, with profound implications in both CRC development and Inflammatory bowel disease (IBD) severity. The complex and dynamic role of glycosylation of mucins in IBD and CRC pathogenesis warrants comprehensive studies to delineate the underlying mechanisms and to develop targeted therapy against mucin glycans.

## 4. Regulation of Colonic Mucin Expression: Normal and Pathological Conditions

As stated above, MUC2 is the most abundantly expressed colonic mucin. Numerous studies have focused on the regulation of this mucin in both pathological and normal conditions. However, in light of the frequently observed aberrant mucin profile in certain colon cancer subtypes, the regulation of other mucins such as MUC5AC has also been studied. In addition, it has been observed that factors such as intestinal microflora and dietary metabolites can also regulate mucin composition in both the human and murine colon.

Commelli et al. observed lower levels of *Muc1* and *Muc4* in the ileum and colon of conventionalized animals with normal microbiota and germ-free mice. Interestingly, a microarray analysis showed that genes regulating intracellular mucin trafficking such as cytoskeletal proteins are significantly altered by the presence of microflora [57]. Wrzosek et al. used a gnotobiotic mouse model to show that the acetate-producing commensal bacterium *Bacteroides thetaiotaomicron* promotes the secretory lineage, i.e., increases the proportion of goblet cells and mucins such as *Muc2* and *Muc4*. Additionally, *B. thetaiotaomicron* also favors the production of sialylated over sulfated mucins. However, the effect of *B. thetaiotaomicron* is countered by another commensal bacterium *Faecalibacterium prausnitzii* that attenuates the effects of *B. thetaiotaomicron* on mucins [24].

Both the diet and microflora play an important role in colonic mucin regulation in the normal gut. In a study with weaned piglets, it was found that dietary zinc can upregulate the percentage of mucin-secreting goblet cells in the colon, concomitant with an increase in the mRNA levels of *MUC1*, *2*, *13*, and *20* [25]. Furthermore, it has been observed that butyrate produced from carbohydrate fermentation by gut microbiota can increase *Muc1* levels through *Muc4* in mice that are given butyrate enemas [58]. Somewhat paradoxically, the thickness of the adherent mucus layer in these mice was reduced. Further, emphasizing the importance of the interplay between diet and microflora, a recent study found that the adherence of probiotics was increased by butyrate treatment, concomitant with an increase in *MUC4, 3,* and *12* in the LS174T human colorectal cancer cell line, reducing *E. coli* adherence as a consequence [26].

In addition to the microbial and dietary influence on colonic mucins, it has been observed that external stressors such as burn injury and alcohol can influence the colonic mucin profile. Hammer et al. found that the combination of alcohol consumption and burn injury caused a significant reduction in colonic *Muc2* and *Muc4* in mice [27]. While the levels of these mucins were normalized 3 days after the insult, *Muc3* and *Muc1* levels were significantly reduced even 3 days after the insult, concomitant with altered levels of *Enterobacteriaceae*. On the other hand, the satiety hormone leptin can also influence levels of colonic mucins. In a study involving both human CRC cell line HT29-MTX and rat cell lines as well as in vivo models, it was observed that leptin stimulates the expressions of MUC2, MUC5AC, and MUC4 in human and of *Muc2, Muc3*, and *Muc4* in rat cell lines via the Protein Kinase C (PKC), Phosphoinositide 3-kinase (PI3K), and MAPK pathways [59]. The antibacterial peptide Cathelicidin can also increase transcription of the *Muc1* and *Muc2* mucins via stimulation of the MAPK pathway [60]. The pathogen *Entamoeba histolytica* was found to stimulate mucin secretion in the LS174T CRC cell line via a PKC-dependent mechanism [61]. Apart from the nonspecific modulators like dietary factors, cytokines, bile acids, and microbiota, specific transcriptional and epigenetic regulations of MUC2, MUC5AC, and MUC4 expression in the colon have been studied extensively. The specific epigenetic regulators for each of these mucins are listed in Table 1, whereas transcriptional regulators and mode of their regulation on mucin gene expression are listed in Table 2.

Other than the mucins mentioned in Table 1 and Table 2, specific regulations of other mucins are also studied, although not as extensively. The transmembrane mucin MUC17 is one of the most prominent membrane-associated mucins in the colon. Like other mucins, it has been observed that a combination of promoter methylation and histone H3-K9 modifications regulate MUC17 expression in CRC. In a study using the MUC17 non-expressing cell line CaCo2 and the MUC17 expressing cell line LS174T, it was observed that methylation occurs at five sites in the promoter-determined MUC17 expression [62]. In addition, it was observed that histone H3-K9 methylation was inhibitory and that acetylation was conducive to MUC17 expression at these CpG sites. Also, a panel of five micro-RNAs (miRNAs) were revealed as putative MUC17 regulators: miR-17, miR-20a, miR-20b, miR-30c, and miR-30e. Like MUC17, the membrane-bound mucin *MUC3A* has also been shown to be regulated by promoter methylation. Here, 30 CpG sites from −660 to +273 in the proximal promoter were found to be critical for MUC3A expression in the LS174T and CaCo2 cell lines [63]. Another abundant transmembrane mucin MUC13 is significantly expressed in metastatic colon cancer tissues. In addition, MUC13 is also expressed at RNA and protein levels in CRC cell lines [64]. Other factors, such as a high protein diet, have been found to influence colonic mucin expression. In a study, rats fed with a high protein (53%) diet displayed an increased goblet cell concentration in the colonic crypt concomitant with decreased goblet cells at the epithelial surface as well as an increase in *Muc3* mucin expression [65].

## 5. Mucin Expression Patterns in Benign and Malignant Conditions

### 5.1. Mucins in Inflammatory Bowel Disease: Ulcerative Colitis and Crohn’s Disease

Inflammatory bowel disease refers to two distinct diseases: Crohn’s disease (CD) and ulcerative colitis (UC). Both these diseases are characterized by a deranged, inappropriate immune response against commensal microflora in the lumen of the gut [66]. Importantly, these diseases confer a strong predisposition towards CRC [67,68,69]. This predisposition is stronger in CD where a 20-fold higher-than-normal risk of CRC was observed [69] in comparison to UC, where the increase in risk is not as significant [68]. In addition, individuals with CD tend to develop mucinous CRCs with a significantly poorer prognosis [69]. These diseases also differ in other respects; CD tends to be transmural (involving the entire gut wall), while UC only involves the mucosa and submucosa [66]. Since the primary function of mucins in the gut is to act as a barrier between intestinal microflora and the surface epithelium, one would expect an alteration in mucin expression in IBD. Studies have shown that there is a more significant depletion of mucins in UC than in CD [70] and that the mucus layer thickness is increased in CD but not UC [71].

### 5.2. Molecular Interplay of Mucins during Colon Pathogenesis

The expression of MUC1 in the normal colon has been associated with homeostatic maintenance. Further, overexpression of MUC1 is observed during inflammatory bowel disease [72]. Gibson et al. observed MUC1 expression in 27% of serrated polyps with no significant difference among different polyp subgroups (HP, SSA/P, and Traditional serrated adenomas (TSA) [73]. Moreover, aberrantly increased expression of MUC1 has been shown in 19–76% of colon adenoma cases along with increased dysplasia [74,75]. Furthermore, overexpression of MUC1 is seen in the vast majority of colorectal tumors with different studies mentioning a variable expression in a range from 30–100% [75,76,77]. Increased MUC1 expression has been demonstrated to be related to increasing TNM or Dukes stage, metastasis, poor tumor differentiation, and worse long-term survival [78,79,80,81]. MUC1 expression was more frequently found in tumors with higher tumor stage and tumor grade of mismatched repair-proficient CRC [82].

MUC4 expression levels were first observed by Ogata and coworkers in eight CRC tumors. However, diminished MUC4 expression was observed in four tumors and the remaining had either an increased expression or were similar to normal tissue [83]. Based on the immunohistochemistry, staining of MUC4 was observed in precursor lesions of CRC and it was reduced in serrated adenoma and in 50% of HP, whereas tubular adenoma (TA) had normal levels of MUC4. In another study, 66% of non-mucinous tumors had low to moderate levels of MUC4, while 34% of CRC had higher MUC4 in it. Patients who had higher expression of MUC4 showed 79% of grade I lesions, and 90% of the lesions which express MUC4 were well or moderately differentiated [84].

The prognostic significance of MUC4 in CRC was assessed in a few studies. In a large cohort of CRC samples, 75% tumors had loss of MUC4, whereas an increase in the MUC4 expression was observed in 25% of tumors. However, its expression in stage I and II patients conferred a worse prognosis [85]. In a recent study, high MUC4 expression with poorer prognosis was observed in 33% of tumors [86]. Therefore, despite the paucity of studies that examine the role of MUC4 in CRC, most studies indicate that MUC4 is lost in the late stages of CRC, though upregulation of MUC4 in the early stages of CRC appears to predict a worse prognosis. It has been proposed that cancer-associated truncated glycan epitopes may cause the altered expression of MUC4 which might be due to altered affinity of antibodies [84]. Moreover, our lab has showed the pro-tumorigenic role of MUC4; when MUC4 knockout mice were subjected to Azoxymethane/DSS treatment, it was observed that the presence of MUC4 led to a significant increase in colitis and colorectal cancer [87].

MUC17 is a membrane-bound mucin that is considered a true structural homolog of rodent Muc3 [88]. Its expression is known to be exclusively high in transverse colon on the surface and crypt columnar epithelial cells [89]. Studies from our lab showed a variable MUC17 expression (mild to intense) in 100% normal colon cases [5]. In contrast to this, an increase in MUC17 (82-fold) expression was observed in SSA/Ps that differentiated from hyperplastic polyps, adenomatous polyps, and normal controls [90].

Gel-forming mucin MUC2 is secreted by goblet and columnar cells of gut epithelium and is organized into two layers [91]. It is a critically important protective effect in the colon and has been demonstrated by utilizing MUC2-null mice that developed spontaneous colitis and adenomas leading to invasive adenocarcinoma [92,93]. Similar to these results, Ajioka et al. found that MUC2 expression was reduced in both flat and polypoid adenomas [94]. Various immunohistochemical studies have reported MUC2 positivity in CRC ranging from 21–63% [95,96], and it was correlated with prolonged survival with low incidence of liver and nodal metastasis [97,98]. Although MUC2 expression is decreased both at the transcript and translation levels of colonic adenocarcinoma, its over or ectopic expression of MUC2 is observed in colonic mucinous carcinomas [96,99,100]. The nature and extent of CpG island methylation have been associated with suppression of MUC2 synthesis [28,101]. In addition, the high mRNA ratio of MUC2/carcinoembryonic antigen (CEA) in CRC patients’ lymph nodes had better prognosis [102].

Ectopic expression of MUC5AC was observed in the inflammatory bowel diseases and in UC-associated dysplasia/neoplasms [103,104]. Investigation of MUC5AC expression in normal-polyp-adenoma-carcinoma sequence revealed no expression of MUC5AC in normal mucosa (NM), while its staining score was higher in HP, TA with low grade dysplasia (LGD) or high grade dysplasia (HGD), mucinous carcinoma (MC), and signet ring cell carcinoma (SRCC) groups compared to NM and colorectal adenocarcinoma groups [105]. Kim et al. investigated the relation of MUC5AC to serrated pathway-associated colorectal cancer and observed a 61% positive rate of MUC5AC expression in colorectal polyps and SSA/Ps and a significant association with sporadic tumors in MSI/H CRC cases [106]. In the cases of SSA/Ps with carcinoma, MUC5AC immuno-expression was observed in 100% of SSA/Ps, in 62.5% of carcinoma, and irregularly in invasive carcinomas [107]. The incidences of MUC5AC expression were 75.4%, 80.4%, and 43.1% in HPs, SSAs, and TSAs, respectively, and concluded that SSAs and HPs each showed significantly higher positive rates for MUC5AC than TSAs [108]. Concurrent to these studies, we did not observe any expression of MUC5AC in normal colon cases while its expression increased significantly in colon adenoma cases in comparison to normal colon and colon inflammation [5]. Overall incidence of MUC5AC was observed to be in 63% adenoma cases. Further, aberrant expression of MUC5AC is observed in proximal, poorly differentiated adenocarcinomas of colon independent of MSI status and in mucinous CRC cases [109,110]. Nakae et al. reported a significantly higher expression of MUC5AC along with high Ki-67 staining and prevalence of KRAS and BRAF mutation in granular-type laterally spreading colorectal tumors [111].

## 6. Mucins in Cell Signaling Pathways: Benign and Cancer

Mucins play a significant role in cell signaling in the normal colon as well as colonic diseases. Figure 3 depicts diverse mucin signaling in benign and carcinogenic events of the colon. In addition to its role in CRC, MUC1 also plays a role in colitis and facilitates the development and progression of IBD and cancer progression. MUC1 was found to play a role in suppressing an inflammatory response involving Th17 signaling. It is believed that loss of MUC1 variants induces excessive Th17 activity, resulting in a deranged inflammatory response characteristic of IBD [112]. A number of studies have also explored the interaction of the chief colonic secretory mucin MUC2 with various signaling pathways in both IBD and cancer. The expression of MUC2 is generally associated with a more differentiated phenotype in CRC. Souaze et al. discovered that expression of the transcription factor Hath1 causes MUC2 and p27 expression, both tumor suppressor genes in certain CRC [113]. In IBD, signaling pathways involving the ER stress response, autophagy, and fatty acid synthase proteins have been shown to alter mucin synthesis and secretion, thus affecting disease severity. Genetic deletion of the unfolded protein response (UPR)-related transcription factor X-box binding protein 1 (XBP-1) led to decreased synthesis of MUC2, consequently increasing susceptibility to IBD [114]. Furthermore, knockout of the ER protein folding gene ARG2 also caused decreased MUC2 secretion. Interestingly, the fatty acid synthase enzyme has been found to be required for the S-palmitoylation of the MUC2 N-terminus and, therefore, is required for proper MUC2 biosynthesis [115]. In our previous study, we showed the role of MUC4 to drive intestinal inflammation and its associated tumorigenesis by using a genetically engineered MUC4 knockout mouse model [87]. To address MUC4 role in colitis and colitis-associated CRC, we used the dextran sodium sulfate (DSS) experimental model and observed that MUC4 knockout mice showed increased resistance to DSS-induced colitis as compared with wild-type (WT) littermates. In addition, as compared to WT animals, MUC4 knockout mice displayed a reduction of inflammatory CD3 (+) lymphocytes infiltration along with cytokines at transcript levels in the inflamed mucosa [87].

MUC1, owing to its interactions with β-catenin, has been the focus of intense investigation in this regard. MUC1 co-localizes with nuclear β-catenin at the invasive front of CRC and this co-expression is associated with a worse prognosis in CRC [116]. In a study performed in the HCT116 CRC cell line, it was observed that MUC1 induces the expression of FOXO3a, a transcription factor involved in DNA repair and oxidant scavenging, via inhibition of the PI3K/AKT pathway. This was theorized to play a role in reducing CRC cell lines’ susceptibility to oxidative stress [117]. MUC1 has also been found to inhibit apoptosis in CRC cells via interaction with JNK1, a member of the mitogen-activated protein kinase (MAPK) family. The MUC1-C was found to interact with JNK1, thus attenuating cisplatin-induced apoptosis [118]. In a study with the HCT116 CRC cell line, MUC1-cytoplasmic domain (MUC1-C) was found to localize to the mitochondria, thereby abrogating cisplatin mediated apoptosis. The mitochondrial targeting of the MUC1-C was found to be triggered by Heregulin, a ligand for Erbb2-4, because both MUC1 and HER2 were found to be co-expressed in stage II and III CRC [119]. In our MUC4 promoter analysis data, we observed three putative TCF/LEF binding sites, suggesting its regulation through Wnt/β-catenin signaling and indicating that reduction of MUC4 at the transcript as well as protein levels occurs, suppressing the Notch pathway effector Hath1 by Wnt/β-catenin pathway in CRC [120]. MUC1 also aids in CRC progression by stimulation of the NF-κB pathway. The MUC1-C promotes NF-κB signaling, which induces the expression of TGFβ-activated kinase1 (TAK1). It also interacts with TAK1 and potentiates the TAK1-TRAF6 interaction, which further activates NF-κB signaling for CRC progression [121]. Another transmembrane mucin, MUC13, which is overexpressed in CRC, has been found to induce the expression of telomerase reverse transcriptase, sonic hedgehog, B cell lymphoma murine like site 1, and GATA-like transcription factor 1 in CRC cells. It has also been found to increase HER2 and phospho-ERK expression in CRC, pointing toward a possible role in aiding CRC progression [122]. Recently, we observed that secretory mucin MUC5AC physically interacts with transmembrane protein CD44, in CRC cells, facilitates cancer progression via Src signaling, and confers chemoresistance through the β-catenin pathway [123].

## 7. Conclusions

Accumulating clinical and epidemiologic evidence over the last few decades has established that mucins are significant molecules governing the pathogenesis of benign and malignant diseases of the colon. Under physiological conditions, mucins maintain epithelial integrity and cellular homeostasis. Differential gene expression and altered glycosylation of mucins are chief drivers of disease outcome, and promoter accessibility and alternate splicing also contribute to this complex interplay. Recent studies emphasize the symbiotic selection of glycans on the mucin backbone and the gut microflora. In addition, transmembrane mucins are extensively studied in the context of pro-tumorigenic signaling while secreted mucins have been in the limelight as a protective double-layered sheet on the colon epithelia against unfavorable environment, toxic metabolites, inflammatory conditions, and microbial infiltration. Efforts have successfully identified the prognostic association of these mucins with CRC disease states; however, the field warrants a comprehensive understanding of the specific domains of the mucins and their interaction involved in disease-associated signaling axes. Since the colon is an ever-renewing organ that sheds off the mucus layer constantly, studying the role of mucins in the context of CRC development and progression will be intriguing as well as rewarding for improved therapeutic targeting of IBD and CRC. Finally, owing to the different role of mucins and their associated glycans in benign colon pathologies versus CRC, current research should be focused on delineating the differential regulation of expression, glycosylation, and signaling partners of these mucins in different disease states.

## Figures and Tables

**Figure 1 cancers-12-00649-f001:**
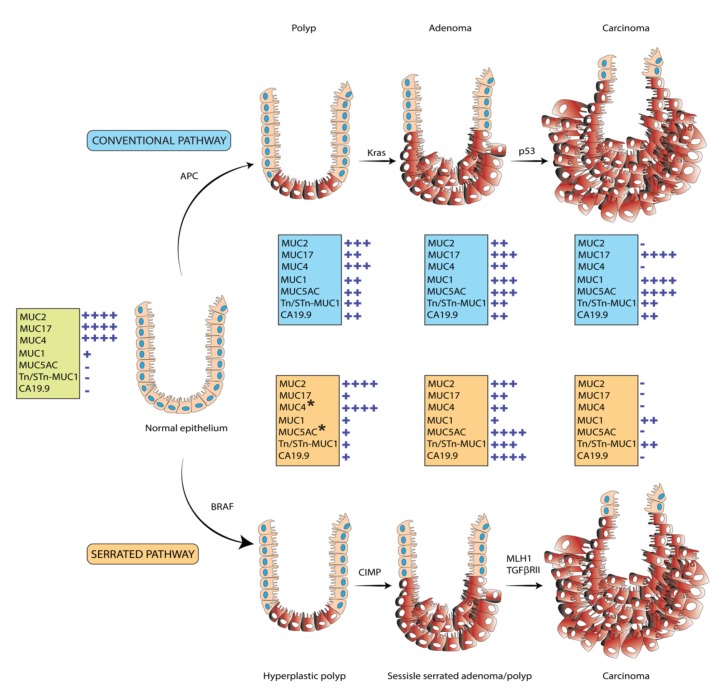
Mucin expression in precursor lesions leading to development of colon cancer. Traditional or conventional development of colon cancer is accompanied by frequent mutations in Adenomatous polyposis coli (APC), Kras and p53 mutations with pathological formation of polyps-adenoma-carcinoma. In contrast, serrated pathway is accompanied by preponderance of BRAF-mutation, high in CpG island methylator phenotype (CIMP) and have MSI-H or microsatellite stability (MSS) along with TGFβRII mutations. Differential expression of mucins and associated O-glycans in polyp-adenoma-carcinoma sequence of both pathways. * Differential localization of mucins (MUC4 and MUC5AC) is observed in the hyperplastic and sessile serrated adenomas/polyps’ subtypes of serrated pathway.

**Figure 2 cancers-12-00649-f002:**
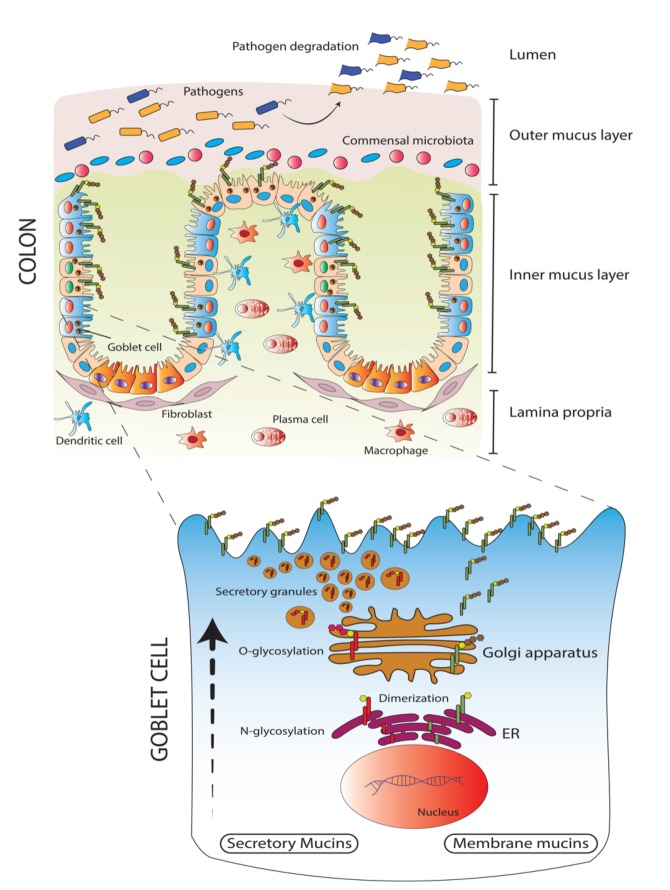
Schematic representing mucin synthesis and localization across colon villi. Transmembrane mucins are localized on the surface of epithelial cells while, secreted mucins (most predominately MUC2) are packed in secretory granules. In presence of low pH, degranulation occurs, and the secreted mucins are released which further undergoes concatenation and forms sheets on the outer layer of the colon epithelium. Depending on the variability of glycans on the secreted mucins, diverse commensal bacteria colonize the outer layer of the colon while the inner layer of the epithelia is impenetrable. Owing to the symbiotic association between MUC2 sheets and commensal bacteria, pathogenic bacteria get expelled; thus, loss of secreted mucins render the inner epithelia vulnerable to pathogenic infiltration. Enlarged is a goblet cell where the membrane-bound and secreted mucins following translation, undergoes post-translational modification in the Endoplasmic Reticulum (ER) and Golgi apparatus. Dimerization and N-glycosylation of the mucins occur in the ER followed by O-glycosylation in the Golgi and then they are either packed in secretory vesicles or targeted to the membrane.

**Figure 3 cancers-12-00649-f003:**
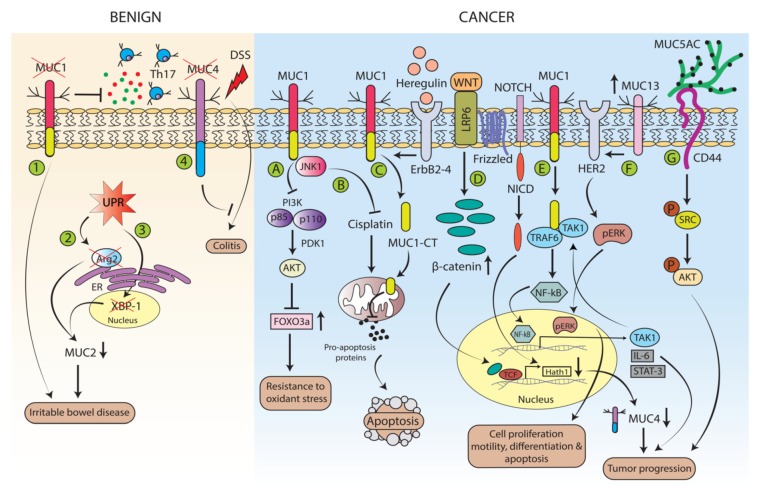
Mucins signaling in the benign and malignant pathologies of the colon. Role of mucins during benign development. (**1**) Transmembrane mucin Muc1 upregulated by Th17 to serve as negative regulator of Th17 signaling. During development of Inflammatory bowel disease (IBD) MUC1 downregulation or its variant expression results in disruption of negative feedback leading to aggravation of IBD. (**2**) Mucin secreting goblet cells are highly dependent upon the unfolded protein response (UPR) pathway for proper functioning. Loss of key component of UPR machinery including Arg2, XBP1 (key protein involved in proper protein folding) leads to reduction in the expression of MUC2 and in turn affecting development of IBD. (**3**) Similarly, downregulation of transcription factor X-box binding protein-1 (XBP-1) in the nucleus results in reduction in MUC2 expression and increases susceptibility to IBD. (**4**) Finally, genetically engineered MUC4 knockout mice show increased resistance to Dextran Sodium Sulfate (DSS) induced colitis. Role of mucins during CRC development. (**A**) MUC1 via PI3K/AKT pathway attenuate phosphorylation of FOXO3 that is involved in inducing oxidative stress in colon cancer cells activity. Overall MUC1 downregulation enhance apoptosis of colon cancer cells. (**B**) The C-terminal domain of MUC1 (MUC1-CT) interacts with JNK1 and attenuate cisplatin-mediated apoptosis. (**C**) Upon induction of ErbB by Heregulin, MUC1-CT is released, and it translocate to the mitochondria to prevent stress-induced apoptosis. (**D**) Binding of Wnt ligand to its receptors i.e., LDL Receptor Related Protein 6 (LRP6) and Frizzled receptor leads to nuclear accumulation of β-catenin which inhibits Hath1 expression via Hes1 (not shown here) along with notch intracellular domain (NICD) results in reduced expression of MUC4 in CRC. (**E**) MUC1-CT interacts with TAK1/TRAF6 complex to induce NFκβ-mediated tumor-promoting gene transcription. (**F**) Over-expression of MUC13 potentiates Her2 signaling and induces ERK-mediated upregulation of genes governing cell proliferation, motility, differentiation and apoptosis. (**G**) Physical interaction of secretory mucin MUC5AC with transmembrane protein CD44 enhanced CRC progression through activation of Src signaling.

**Table 1 cancers-12-00649-t001:** Epigenetic regulation of secreted and transmembrane mucins of colon.

Mucin Type	Epigenetic Regulation	Impact on Expression	Ref.
**MUC2**	Promoter methylation at specific sites	50% promoter de-methylation induces MUC2 expression in CRC cell line. Methylation at specific sites of promoters of non expressing cell lines. Normal goblet cells (express MUC2) have de-methylated MUC2 promoters, while normal columnar epithelium cells have highly methylated promoters.	[28,124,125,126]
	Histone modification	Short chain fatty acids like butyrate and propionate cause methylation and acetylation on histone 3 and histone 4 in MUC2 promoters, leading to its expression in goblet cells.	[29]
**MUC5AC**	DNA methylation	De-methylation at −3718 to −3670 upstream of the transcription start site induce MUC5AC expression in several CRC cell lines. Hypomethylated MUC5AC gene correlates with high expression in precursor lesions, sessile serrated adenomas, and microvascular hyperplastic polyps. MUC5AC hypomethylation is a predictive determinant for microsatellite instability (MSI) CRC tumors.	[30,126,127]
**MUC4**	Promoter methylation	Methylation at five critical residues from −170 to −102 in the 5’UTR is required for MUC4 expression in CRC cell lines except LS174T.	[128]
	Histone acetylation	Treatment with Histone deacetylase (HDAC) inhibitor Trichostatin increased MUC4 expression in low-expressing and non-expressing cell lines but suppressed its expression in MUC4-rich cell lines.	[129]

**Table 2 cancers-12-00649-t002:** Transcriptional regulation of secreted and transmembrane mucins of colon.

Mucin Type	Regulator	Mode/Pathway of Regulation	Ref.
**MUC2**	Galectin-3	Galectin-3 is a ligand of MUC2 and is associated with highly metastatic mucinous colorectal cancer (CRC). Galectin-3 upregulates MUC2 expression via AP-1 transcription factor.	[130,131]
	Bile acids: deoxycholate, chenodeoxycholate, and ursodeoxycholate	Bile acids induce MUC 2 expression via Protein Kinase C (PKC) activation, independent of the MAPK pathway.	[132]
	Secondary bile acid: Deoxycholic acid (DCA)	DCA stimulates MUC2 expression via the EGFR/RAS/MEK1/ERK1/2 pathway, the PKC/p38/MSK1/CREB pathway, and the IKK/IKB/NF-κβ pathway.	[31]
	Intestine specific homeobox transcription factor CDX-2	CDX-2 upregulates MUC2 expression via two sites, −177/−171 and −191/−187 sites, upstream of transcription start sites.	[133]
	Wnt/β-catenin pathway	Wnt/β-catenin represses MUC2 expression via Sox9 transcription factor.	[134]
	GATA-4	GATA-4 upregulates MUC2 expression in murine cell lines.	[135]
	Vasoactive intestinal peptide (VIP)	VIP increases MUC2 expression via ERK and p38 pathways.	[136]
**MUC5AC**	Smad-4 and Sp-1	These transcription factors cooperatively upregulate MUC5AC expression in mouse.	[137]
	GATA-6 and HNF-4α	These transcription factors cooperatively upregulate MUC5AC expression in mouse rectal cell line CMT-93 and thus are believed to play similar regulatory roles in CRC.	[138]
	Sox2	Sox2 transcriptionally upregulates MUC5AC in CRC cell lines, serrated polyps, mucinous, and signet cell carcinomas.	[139]
	Trefoil factor 3 (TFF3)	TFF3-mediated AKT signaling followed by nuclear β-catenin upregulates MUC5AC expression in HT-29 cell line.	[140]
**MUC4**	Transcription factors AP-1, AP-2, Sp1, Sp3, STATs, and GATAs	These transcription factors have binding sites on MUC4 proximal and distal promoters.	[129]
	Developmental transcriptional factors	CDX1, CDX-2, FOXA1, and FOXA2 induce high MUC4 expression in colon cancer cell lines. HNF-1α and HNF-1β induce MUC4 in all cell lines. HNF-4α, HNF-4β, FOXA2, and GATA-5 induce MUC4 in an indirect fashion.	[141,142]

EGFR = epidermal growth factor receptor; HNF = hepatocyte nuclear factor.

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
