# Peer review of "Mechanistic and Functional Shades of Mucins and Associated Glycans in Colon Cancer"

_cancers, 2020, doi:10.3390/cancers12030649_

Round 1
Reviewer 1 Report
Review article; Mechanistic and Functional Shades of Mucins and Associated Glycans in Colon Cancer
Reviewer’s comments;
Overall, the manuscript covers broad information about regulation and function of mucins in normal and pathological conditions, and thus will be informative to wide-range of readers. Notably, they discussed that specific mucin expression is involved in specific CRC subtype, such as sessile serrated adenomas/polyps (SSA/Ps), which is very informative for colon cancer prevention research.
However, it is long and the description is redundant in chapter 2 (line71-195), where authors mainly discuss about innate MUC family composition and expressions in normal colon epithelium. I think that this review article should more strongly discuss about mucins on colon cancer or inflammatory bowel diseases situation. Thus, it would be better to reorganize this chapter to be more compact.
Minor comments;
Line 38: The authors describe “BRAF mutations are the most frequent (70-100%) in CRC that…”. According to TCGA data (Nature 2012), most BRAF mutations are observed in hypermutated MSI CRC, whereas APC is the most mutated gene in non-hypermutated MSS CRC. Thus, the authors should describe clearly as “BRAF mutations are the most frequent (70-100%) in hypermutated MSI CRC that…”.
Line 46-48: In addition to the described classification of CRC, recently accepted concept of molecular subtypes of CRC is Guinney J group’s study (Nature Med. 2015, 21;1350-1356). They divided CRC into four consensus molecular subtypes (CMSs): CMS1; Hypermutated, MSI, strong immune activation. CMS2; Epithelial, marked WNT and MYC signaling activation. CMS3; Epithelial and evident metabolic dysregulation. CMS4; prominent TGFb activation, stromal invasion and angiogenesis. It would be nice to include such new classification in the text.
Figure 1 and Figure 2
The epithelial cells illustration is shown in each figure, however, the apical side (with micro villus) and basal side (with smooth line) are seem to be oppsite between them, is it OK?
Reviewer 2 Report
In this manuscript, Ramesh Pothuraju et al. have reviewed the significance of mucins and associated glycans in colon cancer.
I think the review is comprehensive and covers an important topic. The illustrations are nicely done and serve as helpful visuals of the main points. The manuscript has the potential to make an important contribution to the literature base. I have just a few minor comments to improve the manuscript:
The sentence on lines 46-48 is a bit difficult to understand. Please consider reformulating.
The sentence on lines 54-55 (Fig. 1 legend) is also a bit unclear.
The statement on lines 65-66 (“Despite being highly overexpressed in most cancers, mucin expression is well regulated by a plethora of direct and indirect factors”) would need a reference to support it.
The sentence on lines 331-333 may be a bit misleading, as TVA and TA are not subgroups of serrated polyps.
